# The Underlying Nature of Epigenetic Variation: Origin, Establishment, and Regulatory Function of Plant Epialleles

**DOI:** 10.3390/ijms22168618

**Published:** 2021-08-10

**Authors:** Thanvi Srikant, Anjar Tri Wibowo

**Affiliations:** 1Department of Molecular Biology, Max Planck Institute for Developmental Biology, 72076 Tubingen, Germany; thanvi.srikant@tuebingen.mpg.de; 2Department of Biology, Faculty of Science and Technology, Kampus C, Airlangga University, Mulyorejo, Surabaya 60115, Indonesia; 3Biotechnology of Tropical Medicinal Plants Research Group, Kampus C, Airlangga University, Mulyorejo, Surabaya 60115, Indonesia

**Keywords:** epiallele, DNA methylation, gene regulation, chromatin, transposable elements, transgenerational inheritance, agricultural innovation

## Abstract

In plants, the gene expression and associated phenotypes can be modulated by dynamic changes in DNA methylation, occasionally being fixed in certain genomic loci and inherited stably as epialleles. Epiallelic variations in a population can occur as methylation changes at an individual cytosine position, methylation changes within a stretch of genomic regions, and chromatin changes in certain loci. Here, we focus on methylated regions, since it is unclear whether variations at individual methylated cytosines can serve any regulatory function, and the evidence for heritable chromatin changes independent of genetic changes is limited. While DNA methylation is known to affect and regulate wide arrays of plant phenotypes, most epialleles in the form of methylated regions have not been assigned any biological function. Here, we review how epialleles can be established in plants, serve a regulatory function, and are involved in adaptive processes. Recent studies suggest that most epialleles occur as byproducts of genetic variations, mainly from structural variants and Transposable Element (TE) activation. Nevertheless, epialleles that occur spontaneously independent of any genetic variations have also been described across different plant species. Here, we discuss how epialleles that are dependent and independent of genetic architecture are stabilized in the plant genome and how methylation can regulate a transcription relative to its genomic location.

## 1. Introduction

DNA methylation is an epigenetic mark conserved across various kingdoms in the tree of life. Although different nucleotides can be methylated, the most common form of DNA methylation in higher organisms is cytosine methylation. It occurs through the covalent addition of a methyl group at the fifth carbon of cytosine, forming 5-methylcytosine (5mC). In plants, cytosine methylation can occur in different sequence contexts: the symmetrical CG and CHG and asymmetrical CHH contexts (where H is A, C, or T). The distribution of methylation in these three contexts can be diverse across plant species. In the CG context, for example, the methylation levels can range from 5.4% in green algae (*Chlamydomonas reinhardtii*) to 58.4% in the rice (*Oryza sativa*) genome [1], reflecting distinct epigenetic regulatory mechanisms in each genome.

Plant phenotype and underlying gene expression profiles can be modulated through dynamic changes in the chromatin properties and DNA methylation [2]. These changes can arise as epialleles, epigenetic variations that are fixed at certain loci and inherited stably across generations. In plants, epialleles are often discovered among genomic regions with differential methylation (called Differentially Methylated Regions or DMRs) [3]. Some DMRs are known to affect the expression of overlapping, proximal, or distantly located genes, making them an important source of variations and phenotypic plasticity, especially in the absence of genetic variations [4]. However, there are also numerous DMRs identified between various plant genotypes that have not been assigned any biological function, despite their proximity to genes.

In the model plant *Arabidopsis thaliana*, the methylation changes at single cytosine positions can occur spontaneously across generations [5,6,7]. Nevertheless, these spontaneous epimutations rarely form an epiallele, since they occur randomly, and a majority of the methylation patterns in the *A. thaliana* genome are inherited faithfully during sexual reproduction [8]. Despite the rare occurrence of spontaneous epialleles, DNA methylation is highly variable at the genomic level, either between plant species or between different populations of plants in the same species [9,10]. Many naturally occurring epialleles with diverse biological roles have been identified in *A. thaliana* [11,12,13,14,15,16,17,18], tomatoes (*Solanum lycopersicum*) [19,20], melons (*Cucumis melo*) [21], rice [22,23], *Linaria vulgaris* [24], and maize (*Zea mays*) [25,26]. While these methylation marks and epiallelic variations found in natural populations are mainly attributed to genetic variations in epigenetic machinery and transposable element (TE) insertions, there are nevertheless variations in DNA methylation that occur regardless of any genetic variation.

Besides naturally occurring in the population, epialleles can also be artificially induced by a wide range of experimental stimuli and treatments. They might arise following biotic or abiotic stress [4], chemical treatments [27,28], clonal propagations [29], interference of the methylation pathways [30,31], and/or due to deficiencies in the maintenance of their chromatin states [31,32]. Considerable evidence shows that induced epimutations can be fixed and stably inherited across generations to form epialleles and, in some cases, may contribute to novel transcriptional regulation and alterations of a plant phenotype [4].

Why some epimutations may be retained and established as epialleles while others are not is a central question in epigenetics. In this review, we will discuss different mechanisms and genomic features that can facilitate the establishment of functional epialleles (epialleles with an assigned biological role) in the plant genome, how they can be retained at a certain locus and stably inherited across generations. Depending on their origin and position in the genome, both naturally occurring and artificially induced epialleles can negatively or positively modulate gene expressions through diverse mechanisms. Here, we will also discuss the recent findings and various mechanistic models for epiallele regulation on gene expression and TE activity, including proteins and small molecules that are involved in the process. Furthermore, we will examine recent technological advances that can be applied to introduce stable and functional novel epialleles in plants, especially tools that can improve plant breeding.

## 2. Naturally Occurring Epialleles

### 2.1. Naturally Occurring Epialleles and Their Biological Functions

Heritable changes of methylation at single cytosine positions can arise spontaneously in plants. A methylome analysis for *A. thaliana* mutation accumulation (MA) lines grown for 30 generations revealed that epimutations occur at a much higher rate compared to genetic mutations (∼10–4 compared to ∼10–9 per sexual generation) [5,6,7,33]. It was proposed that these epimutations are stable enough to effectively respond to long-term selections. A more recent study of the same line confirmed that spontaneous methylation changes can lead to the formation of stable and heritable epialleles (“spontaneous epialleles”), although they occur at a very low frequency. In each sexual generation, spontaneous epialleles that manifested as DMRs constitute less than 0.003% of the methylome, while 99.998% of methylated regions are stable and inherited faithfully in the progeny [8]. In addition, the biological significance of these spontaneous epimutations and epialleles is still unclear, since their role in gene regulation and the establishment of heritable phenotypes independent of genetic variations has not been defined [34].

Although spontaneous epialleles are very rare, DNA methylation is highly diverse across different the natural strains of *A. thaliana*. A methylome analysis of 1107 *A. thaliana* accessions highlighted this diversity, where 78% of the methylated sites in the *Arabidopsis* genome were highly variable, with 22,060 DMRs identified across all the accessions, showing the abundance of naturally occurring epialleles at the intraspecies level [9]. The methylation variants found in these accessions were strongly correlated with their geographical origins and climates, indicating their potential role in the environmental response and adaptation [9]. In accordance, various works have reported their diverse biological functions, including the control of leaf senescence [11], starch metabolism [12], tryptophan biosynthesis [13], hybrid incompatibility [14], resistance to the plant pathogen *Plasmodiophora brassica* [15], and flower development [18]. These natural epigenetic variants are also responsive to selection, as demonstrated by Schmid et al. (2018), who showed that the seed dispersal selection for five consecutive generations of 19 *A. thaliana* recombinant inbred lines (RILs) led to a reduced epigenetic diversity, global methylation changes, and the selection of specific epialleles in the population. The selected epigenetic variation was associated with an altered flowering time, number of branches, and number of siliques [35]. The epigenetic diversity and functional natural epialleles have also been well-described in other species, although the extent has not been as well-explored as in *A. thaliana*. Variations in the methylation patterns have also been identified across 83 soybean RILs [36] and across different wild soybean, landraces, and cultivars [37]. Similarly, thousands of DMRs were also observed in different genotypes and RILs of maize [26,38,39,40] and in the model cereal *Brachypodium distachyon* RILs [41]. As found in *A. thaliana*, naturally occurring epialleles also serve diverse biological functions in other plant species, a classical example being epiallelic and transcriptional variations in the *Lcyc* gene in *L. vulgaris* that controls floral symmetry and the *pericarp color1* (*p1*) locus in maize, which controls the kernel development and pigmentation [24,25]. Such epialleles are also involved in the regulation of the ripening and vitamin E accumulation in the fruits of tomatoes [19,20], sex determination and development of flowers in melons [21], and plant architecture and photosynthesis capacity in rice [22,23].

Natural epialleles can also be generated during reproductive development in a process called imprinting. Notable examples include the *FLOWERING WAGENINGEN* (*FWA*), *MEDEA* (*MEA*), *FERTILISATION INDEPENDENT SEED DEVELOPMENT 2* (*FIS2*), and *HOMEDOMAIN GLABROUS 3* (*HDG3*) epialleles that are transcriptionally regulated by DNA methylation changes in endosperm tissues. The parental allele-specific transcription of these genes is tightly regulated, and an imbalanced expression can result in strong developmental phenotypes, such as delayed flowering and altered seed sizes [17,42].

### 2.2. The Establishment of Naturally Occurring Epialleles

Although the spontaneous epimutation rate at a single cytosine position is much higher than its genetic mutation rate, the significance of this epimutation in shaping biologically meaningful DMRs and epigenomic diversity within a species is unclear, since the majority of methylated regions are conserved and inherited faithfully across generations [8,34]. Instead, the evidence for linkage and a strong association between methylation and genetic variation have been observed in inbred lines of soybean [36], *B. distachyon* [41,43], maize [39], and *A. thaliana* natural populations [9,44,45,46], suggesting that the majority of methylation variations found within a species could have a genetic basis. The genetic control of methylation variations arises mainly from sequence variability of the genes regulating methylation and histone deposition but can also originate from variations in repeat regions and TE transposition [9,44,45,46,47,48,49,50,51].

In *A. thaliana*, the variation in non-CG methylation is strongly associated with the genetic variations of genes involved in the RNA-directed DNA methylation (RdDM) pathway, such as *CHROMOMETHYLASE 2 (CMT 2)*, *ARGONAUTE 1* and *2 (AGO1* and *AGO2)*, and *NUCLEAR RNA POLYMERASE D1B (NRPD1B)* [9,44,45,46], while the variation at the CG gene body methylation (gbM) is associated with the variation in *METHYLTRANSFERASE 1 (MET1)* [9]. The presence or absence of TEs and repeat-rich sequences is also responsible for the observed epigenomic diversity in *A. thaliana* (Figure 1C). TE variants are strongly associated with the methylation level of overlapping or nearby regions in a distance-dependent manner, suggesting the cis-regulation of TE insertions in methylation [46,47]. A more recent study revealed that the TE mobilization rates across different *A. thaliana* accessions are associated with the *NUCLEAR RNA POLYMERASE E1 (NRPE1)* genetic variation. *NRPE1* encodes the largest subunit of RNA Pol V, a main component of RdDM and non-CG methylation. This work suggests that non-CG variations across the *Arabidopsis* accessions are strongly shaped by the *NRPE1* genetic variation and TEs transposition [49]. The effects of TEs on the overlapped or flanking methylated regions are also observed in maize [50] and rice [51], where the insertion of younger TEs in unmethylated regions can cause increased methylation around the insertion site and facilitate the establishment of novel epialleles. In contrast to what is observed in other species, in *B. distachyon*, novel TE insertions or deletions have a limited impact on the methylation level of the nearby regions, and only few TE families were found associated with the nearby gene expression, suggesting that the effect of TEs on methylation could be dependent on the genome structure of the species [43].

In a comprehensive analysis, Zhang et al. (2020) suggested that the epigenetic variation at gbM likely arises from a combination of genetic and/or environmental factors that influence the DNA methylation homeostasis between heterochromatin and euchromatin. This homeostasis is affected by variations in the gene length, CWG content in the genes (where W = A or T), and the gene proximity to TEs or pericentromeric heterochromatin. The authors observed that such a balance in the methylation levels is also governed by the accuracy and efficiency of methyltransferases and chromatin remodelers in a particular genotype and claimed that all of these factors together with the environmental stimuli can shape the epigenomic states of individuals within populations [50]. For instance, they found that plant genotypes with lower methylation levels at heterochromatic regions would have a higher number of genes with CG gbM. Eventually, these genes could be more prone to become methylated at non-CG sites, resulting in their silencing [52].

In imprinted genes, epiallelic regulation occurs in the endosperm by the differential silencing/activation of the maternal or paternal alleles. Maternally imprinted genes (MEGs) such as *FWA, MEA*, and *FIS2* are hypomethylated in specific regions of the maternal allele by the DME glycosylase, thereby activating its transcription (Figure 1A). Paternal imprinting, on the other hand, involves a POLYCOMB REPRESSIVE COMPLEX 2 (PRC2)-mediated deposition of repressive histone marks (H3K27me3) in the paternal alleles, leading to silencing [42]. Although the endosperm is a terminally differentiating tissue and may not directly transmit these epigenetic states to the progeny, several imprinted epialleles occur in the loci carrying master regulatory genes or transcription factors (the class IV HD-ZIP family, for example). Natural variations in the methylation state of these epialleles have been shown to impact endosperm cellularization and seed development [17].

Although a significant portion of epigenomic diversity seems to be linked with genetic variations, several studies have also reported the occurrence of “pure epialleles”, epialleles that arise independent of any genetic variants (Figure 1A) [53]. Differential methylation patterns that do not show evidence for linkages with genomic regions were identified in soybean RILs [36,37] and in maize inbred lines [26]. In *A. thaliana*, the *QUA-QUINE STARCH (QQS)* gene is a good example of a functional pure epiallele. The methylation state and the expression of this gene varies across the *A. thaliana* accessions independent of the genetic variation. This epiallele was identified in a laboratory stock of the reference strain Col-0 due to a spontaneous epimutation event. In inbred lines of the Col-0 and Cvi-0 strains, the hypomethylated variants of *QQS* were stably inherited across generations independent of the genetic variation [12]. Another example of a pure epiallele is epimutation at the tomato Colorless non-ripening *(Cnr)* locus that results in colorless fruits and decreased cell-to-cell adhesion. Spontaneous methylation in the promoter of a *SQUAMOSA PROMOTER BINDING PROTEIN* (*SBP-box*) gene residing in the locus, without an alteration in the DNA sequence, caused the transcriptional repression of the gene and altered the fruit phenotype [19,20]. The exact mechanism for the establishment of these pure epialleles is not fully understood. They might arise due to stochastic errors in DNA methylation maintenance during replication and/or errors during epigenetic reprogramming in germ cells, followed by a subsequent reinforcement in the embryos and differentiated cells. In nature, these spontaneous epimutations might be subjected to long-term selection by various environmental conditions, then fixed as heritable and sometimes functional epialleles (Figure 1A).

## 3. The Origins and Functional Roles of Induced Epialleles

### 3.1. Epialleles Formed in Epigenetic Recombinant Inbred Lines (epiRILs)

Novel epigenomic states can be created by the hybridization of wild-type (WT) and hypomethylated genomes, generating progeny called epigenetic recombinant inbred lines (epiRILs) [53]. In *A. thaliana*, epiRILs can be derived from the cross between a near-isogenic WT plant with *met1–3* [30] or *ddm1* and *ddm2* [31] mutant lines (Figure 2A). *MET1* encodes the main DNA methyltransferase, maintaining CG methylation in both the euchromatic and heterochromatic regions [54], while *DECREASED DNA METHYLATION 1 (DDM1)* encodes a chromatin remodeler protein involved in histone methylation and the maintenance of DNA methylation in all contexts (CG, CHG, and CHH), especially at TEs and repeat sequences in the heterochromatic regions [55]. To generate epiRILs, F2 plants with homozygous *MET1* and *DDM1* WT alleles are selfed for 5–16 generations (Figure 2B) [20,31,56]. A methylome analysis on these lines showed a partial but stable inheritance of the *met1–3* or *ddm1* and *ddm2* hypomethylated regions in the mutant allele-free progenies. Similar to the parental methylomes, the epialleles generated in *met1–3* epiRILs were distributed in both the gene bodies and repeats at the heterochromatin [57,58], while epialleles in the *ddm1* and *2* epiRILs were mainly found at the repeats and heterochromatic TEs [59]. A more recent work on *ddm1* and *2* epiRILs successfully mapped the epigenetic quantitative trait loci (epiQTL) in the segregating epiRILs population and identified a set of heritable DMRs correlated with the flowering time [59], primary root length [59], tolerance to salinity [60], resistance to clubroot infection by *Plasmodiophora brassicae* [15], and resistance to oomycete *Hyaloperonospora arabidopsidis (Hpa)* (Figure 2C) [61].

While epiRIL lines can stably inherit some hypomethylated regions from the mutant parental line, other regions can be remethylated to the wild-type state. In *ddm1* and *ddm2* epiRILs, such remethylation can occur progressively across one to three generations after the F1 hybrid is selfed or reciprocally backcrossed with the WT to produce individuals that lack the mutant allele (DDM1/DDM1). These remethylation events are guided by small-RNAs and the associated RdDM pathways [62]. In *met1–3* epiRILs, remethylation is efficiently carried out in the F1 hybrids, depending on the DNA sequence properties of the region [63]. In general, two factors have been shown to determine the methylation state of the affected region: a low CG content and high repeat copy numbers in a region are associated with remethylation to a wild-type state, while a high CG content and low repeat copy number are associated with the establishment of stable and heritable hypomethylated regions [58] (Figure 2D).

### 3.2. Environmental Stress Can Induce Epiallele Formation

Various biotic and abiotic stresses, including *Pseudomonas syringae* infection [64,65,66], cyst nematode infection [67,68,69], symbiotic bacteria [70], caterpillar herbivory [71], grazing by mammals [72], high salinity [73,74,75], drought [76,77,78,79,80,81], cold [82,83,84], phosphate starvation [85,86], heat [84,87], anoxia and reoxygenation [88], UV light [84,89], excess light [90,91], and iron deficiency [92], have been applied to different plant species and reported to cause a wide range of epigenetic changes. Here, we will focus our discussion on stress-induced epimutations and epialleles with regulatory functions on the gene expression and, consequently, plant phenotype.

In the wild, plants are exposed to pests and predators of all scales, especially to microorganisms in the soil. Exposing plants to bacterial infections under controlled conditions can help us to understand the transient epigenetic changes during infections. For example, a treatment with flg22, an immune-response-inducing fragment of the bacterial flagellin, can cause the hypomethylation of helitron-derived repeats lying within the promoter of the defense gene *RESISTANCE METHYLATED GENE 1 (RMG1)*, ensuing its activation [93]. A *Pseudomonas syringae* pathovar tomato DC3000 *(Pst)* infection can also prime *A. thaliana* to activate salicylic acid (SA)-inducible genes, enhancing the resistance to *Hpa* and *Pst* infections [4,65,66]. This acquired resistance can be inherited in the non-stressed progeny and appears to be stronger when the parental lines are exposed to *Pst* for multiple generations. Although CG DMRs at a few defense response genes were found to be heritable in the non-stressed progenies [65], it is not explored whether these epimutations can directly trigger defense pathways.

Similar observations have been made when *A. thaliana* plants are subjected to an abiotic stress of a high-salt concentration. Multigenerational exposure to salt stress also leads to an increased tolerance in the direct progeny of the stressed plants and is lost in the subsequent generations of non-stressed plants [74]. This transient tolerance is also accompanied with heritable changes in non-CG methylation. A regulatory role of salt stress-induced DMR in the gene expression was demonstrated for one locus harboring differential methylation at a TE downstream of the *CARBON/NITROGEN INSENSITIVE 1* (*CNI1*) gene [74]. Nevertheless, it is unknown whether other salt stress-induced DMRs can serve similar function and are responsible for the increased tolerance observed in the progeny. Besides stochastic genome-wide changes, salt stress is also reported to induce targeted CHH methylation changes at the promoter of *Arabidopsis MYB DOMAIN PROTEIN 74* (*AtMYB74*). In the absence of stress, this region is heavily methylated by the sRNA-guided RdDM pathway. Upon salt stress, fewer sRNAs accumulate, and the CHH methylation levels decrease, thus triggering the upregulation of the gene [94]. The loss of non-CG methylation is also observed in the promoter of *SUPPRESSOR OF DRM1 DRM2 CMT3* (*SDC*) following the exposure to heat stress, causing the activation of this silenced and imprinted gene in *Arabidopsis* vegetative tissue [95]. The effects of drought stress, on the other hand, can vary widely across species. In *A. thaliana*, for example, a drought only induces limited intergenerational methylation and phenotypic changes [77,79]. However, the drought stress for 11 generations is reported to induce stable, heritable, and targeted methylation changes in rice. These drought-induced DMRs directly overlap with several known drought-responsive genes that are differentially expressed in the non-stressed offspring [81]. While there is plenty of evidence to demonstrate that plant epigenomes can dynamically change in response to external stimuli, only a handful of studies reported that these changes can be stably inherited in the absence of external stimuli (Figure 3). Further, evidence for a direct causal relationship is rare, as there is yet no evidence for heritable acquired traits that are exclusively dependent and causally linked to stress-induced epigenetic changes.

### 3.3. Epiallele Formation upon Clonal Propagation

Somatic clones derived from the same plant tissue (explant) are expected to be phenotypically, genetically, and epigenetically identical. However, heritable variations at the phenotypic and molecular levels are often observed in the clonal progenies generated through hormone-induced tissue cultures or by the ectopic expression of embryonic genes in the somatic tissues [96,97,98,99]. Epimutations induced by regeneration have been reported in clonally propagated *A. thaliana* [100,101], rice [102], oil palms (*Elaeis guineensis*) [103], pineapple (*Ananas comosus*) [104], and maize [105,106,107], some of which can be stably inherited as epialleles in the progeny, even after several rounds of sexual propagation.

In maize, regeneration by a tissue culture causes a partial-to-complete silencing of *PERICARP COLOR1 (P1)* expression, causing a reduction or loss of pigmentation in cob glumes. *P1* silencing was associated with hypermethylation in the second intron of the gene [107]. In oil palms, a tissue culture can lead to the formation of a *Bad Karma* epiallele that causes sterile flowers, abortive fruits, and lower oil yields. This epiallele is associated with the loss of CHG methylation at a LINE retrotransposon nested in the intron of the *DEFICIENS* gene, resulting in alternative splicing and premature termination of the gene transcript [103]. Novel epiallele formations during clonal propagation often lead to deleterious phenotypes; therefore, limiting epigenetic variations during clonal propagation is desirable for breeding purposes. Wibowo et al. (2018) reported that the epimutation rates in clonal lines are affected by the meristematic identity of the tissue used for regeneration. Plants regenerated from roots inherit many aspects of root-specific DNA methylation and gene expression patterns, even in their leaves. These epimutations can affect how the clonal plant interacts with the symbiotic and pathogenic microbes. On the other hand, plants regenerated from leaves only showed minor variations in the gene expression and methylation patterns, being phenotypically indistinguishable from normal plants. These findings suggest that epigenetic and phenotypic variations can be modulated merely by choosing the appropriate explant [100].

### 3.4. Epialleles Introduced by Epimutagenesis

One of the early approaches to study the consequences of hypomethylation was the chemical treatment of plants with 5′azacytidine, which functions as an inhibitor of DNA methyltransferases [108]. This chemical treatment has been shown to induce both epigenetic and phenotypic changes in several plant species, some of which are heritable [109,110,111]. With the growing pace of genetic engineering advances, new epimutations and epialleles can currently be introduced into the genome at the locus-specific or genome-wide level. In *A. thaliana*, widespread hypomethylation can be generated through the ectopic overexpression of a human *TEN-ELEVEN TRANSLOCATION 1 (TET1)* demethylase gene. The overexpression of this gene results in a global loss of methylation in *A. thaliana*, especially in the CG context, resembling the molecular phenotype seen in *met1* mutant lines (Figure 4E). Some of TET-mediated hypomethylated DMRs can be stably inherited in the progeny, even after the transgene expressing TET1 is segregated away. In addition, the transgenerational inheritance of hypomethylated epialleles was observed to be higher when the transgene was expressed in meristematic tissue. A causal relationship between TET1-introduced DMRs with the gene expression and plant phenotype was demonstrated for the *FWA* locus. *TET1* overexpression caused a complete loss of DNA methylation in the promoter of *FWA*, resulting in activation of the gene and delayed flowering [112].

DNA methylation can also be introduced at specific sites in the plant genome. Molecular tools known to efficiently target epimutagenesis include Inverted Repeat-Hairpins (IR-Hairpins), zinc finger proteins (ZFs), and the dCas9 SunTag system [113] (Figure 4). Inverted repeats carrying a target sequence can be introduced in the plant genome as transgenic IR-hairpin constructs. The transcription of this sequence within the plant cell results in the formation of hairpin-shaped structures, which are cleaved into siRNAs. These IR-derived siRNAs can trigger the deposition of methylation marks in *trans*, thereby silencing the homologous targets through the RdDM pathway [114] (Figure 4A). This tool has been successfully used in different plant species to introduce methylation at the regulatory elements of various genes [115]. In *A. thaliana*, IR-hairpin constructs can efficiently induce methylation at the promoter of *FWA* [116], *REPRESSOR OF SILENCING 1 (ROS1)* [117], and distal enhancer region of *FLOWERING LOCUS T (FT)* [118], leading to the repression or complete silencing of the targeted genes.

In another approach, engineered Zinc Finger (ZF) arrays can be fused with different RdDM factors to introduce DNA methylation at specific sequences (Figure 4B,C). Various fusions of ZFs with RdDM proteins, including NRPD1, RDR2, SHH1, DMS3, RDM1, DRM 2, SUVH9, MORC6, MORC1, and, recently, MORC7, have been used to introduce methylation at the hypomethylated promoter of *FWA* in the *A. thaliana*
*fwa* epimutant and restore the early flowering phenotype [119,120]. ZFs can also be fused with the catalytic domain of TET1 to induce a targeted demethylation of the *FWA* promoter, resulting in activation of the gene and delayed flowering [121]. Similar to ZFs, the dCas9 protein can also be fused with the catalytic domain of a methyltransferase or demethylase enzyme to induce the targeted addition or removal of DNA methylation. The efficiency of the system can be improved when the dCas9 protein is combined with the SunTag system that allows for the recruitment and multimerization of several protein effectors in the target locus [122]. In *A. thaliana*, a dCas9-SunTag construct can effectively recruit the tobacco DOMAINS REARRANGED METHYLTRANSFERASE 2 (DRM2) catalytic domain into the promoter of the *fwa* epiallelic mutant, introducing methylation to silence the activated gene [123]. The fusion of dCas9 with a bacterial methyltransferase MQ1 (SssI), from *Mollicutes spiroplasma*, was recently shown to be able to introduce CG-specific methylation at the target loci, demonstrating the precision in which methylation marks can be artificially introduced [124]. dCas9 can also be fused with the TET1 catalytic domain to remove methylation at the *FWA* promoter, triggering the activation of the gene and delay in flowering [121] (Figure 4D).

All of these studies have demonstrated that stable and functional epialleles can be artificially introduced in plants. In random epimutagenesis approaches, the introduced epialleles and the various phenotypes accompanying it can be subjected to selection for breeding purposes. While using targeted epimutagenesis, the epigenetic state and transcriptional level of certain genes can be modulated to produce desired phenotypes.

### 3.5. Establishment and Maintenance of Induced Epialleles

The majority of the methylation changes induced by environmental stresses, clonal propagation, or artificial epimutagenesis are reset to the basal levels in the non-stressed, sexual, and transgene-free progeny. However, some changes can be stably transmitted through many rounds of mitotic and meiotic divisions and inherited in the progenies [4,125]. It remains unclear why certain methylation marks are erased during reprogramming events in gametes, while some others can bypass this reprogramming and pass down to the next generation.

A screening for *A. thaliana* mutants impaired in the resetting of stress-induced release of epigenetic silencing identified the roles of *DDM1* and *MORPHEUS MOLECULE 1 (MOM1)* in epigenetic resetting. In *ddm1/mom1* double mutants, heat stress exposure released the methylation-induced silencing of the TEs, transgenerationally carrying over these marks in the non-stressed progeny. However, in single mutants of *ddm1* or *mom1*, this stress-induced activation was reset in the progeny, suggesting their redundant role in the erasure of stress memory. It is suggested that this resetting may require an interplay between DNA methylation, modifications of the chromatin proteins, and nucleosome occupancy, involving many other factors besides *DDM1* and *MOM1* [126]. Recent studies have suggested that chromatin and the DNA sequence properties of a region are the determining factors for the stability of the methylation changes in the region [127]. Following the genome-wide loss of methylation in the *nrpd1* mutant and subsequent reintroduction of the *NRPD1* gene, a low level of CG content in a region is associated with the recruitment and remethylation by RdDM machinery, while regions with a high CG content could retain their hypomethylated state for generations. The stability of RdDM-guided methylation is also dependent on the level of histone modification marks H3K4me3 and H3K18a in the region. High levels of H3K4me3 in a region can restrict the recruitment of RdDM machinery and prevent remethylation, while the high presence of H3K18ac marks can attract ROS1-mediated demethylation, which antagonizes the local RdDM activity [127].

## 4. Regulation of Gene Expressions by Epialleles

### 4.1. Regulatory Function of Epialleles in Proximity to Genes

Changes in methylation are often found upstream of the gene body, either in the promoter, 5′ untranslated region (5-UTR), or the transcription start site (TSS) of the associated gene. Increases in methylation and transcriptional downregulation or silencing are especially common in repetitive sequences and when TEs are found inserted in the regions [128] (Figure 5B). In *A. thaliana*, DNA methylation with such regulatory features has been reported for methylation covering the promoter of *PHEOPHYTIN PHEOPHORBIDE HYDROLASE (PPH)* [11], *HISTIDINE BIOSYNTHESIS 6B (HISN6B)* [129], *RMG1* [93], and *QQS* [12]; methylation at the 5′-UTR region of *FOLATE TRANSPORTER (FOLT)* [130]; and methylation around the TSS region of *FWA* [131]. DNA methylation with a similar repressive function is also found in the promoter of *SBP-box* and *VITAMIN E 3 (VTE3)* in tomatoes [19,20], *OsAK1* [22] and *DWARF1* [23] in rice, and *CmWIP1* in melons [21]. The regulatory effect of promoter methylation in *A. thaliana* may also be sequence- or region-specific; this has been shown in *FWA*, where methylation at tandem repeats overlapping the TSS can completely silence the gene but only has minimal effects on the transcription when found in upstream regions [116].

Methylation can also repress the transcription by directly inhibiting the transcription factors (TFs) binding to the regulatory regions and/or by promoting the recruitment of repressive histone marks and the removal of permissive histone marks, thus restricting protein access into the methylated region (Figure 5B). In *A. thaliana*, around 76% of TFs are sensitive to methylation, most of them (72%) showing a direct binding inhibition by methylation, while 4.3% display a binding affinity to methylated sites, suggesting a complex and sequence-specific effect of DNA methylation on TF binding [132]. Besides its direct role in the inhibition of TF binding, DNA methylation can regulate transcription through dynamic interactions with various histone marks. Examining the chromatin states of various DNA methylation mutants has resulted in a general consensus that DNA methylation is associated with an increased level of repressive histone marks, such as H3K9me2 and H3K27me3, and reduced level of permissive histone marks, such as H3K9Ac and H3K4me3 [55,133,134,135,136]. This correlation is shown in the interaction between CHROMOMETHYLASE 3 (CMT3) and H3K9me2, since CMT3 can bind to H3K9me2 to establish methylation at the CHG sites adjacent to the histone mark. The methylated DNA sequence attracts the histone methyltransferases SUVH4, SUVH5, and SUVH6, which can deposit H3K9 di-methylation around the CHG site, establishing a CHG-H3K9me2 reinforcing feedback loop [137,138]. Under biotic stress, the CHG methylation and H3K9me2 deposition at the defense genes are dynamically regulated by *INCREASE IN BONSAI METHYLATION* (*IBM1*) [139]. In maize, the CHG methylation and H3K9me2 deposition near the TSS were associated with lower transcriptional rates of the genes [134,140]. DNA methylation can also affect the binding of REF6, a Jumonji N/C domain-containing histone demethylase that regulates H3K27me3 demethylation. REF6 recognizes a CTCTGYTY motif via its zinc-finger (ZF) domains, and a high level of non-CG methylation in the motifs prevents RELATIVE OF EARLY FLOWERING 6 (REF6) targeting, facilitating the stable deposition of H3K27me3-repressive marks in the TEs or genes [141]. Besides promoting repressive histone marks, DNA methylation is also known to be negatively correlated with permissive histone marks [142,143]. In rice, reduced methylation in the promoter and gene body are associated with an elevated H3K4me3 and increased transcription. In clonal rice lines, the regeneration-induced loss of methylation at the promoter and 5′-UTR of certain genes is accompanied by elevated levels of H3K4me3. The removal of H3K4me3 by histone demethylase results in an increased level of DNA methylation in the locus and gene repression, suggesting a negative correlation between DNA methylation with H3K4me3 and gene activation [144].

In some cases, DNA methylation may also be required for gene activation, as shown in the regulation of the *ROS1* demethylase in *A. thaliana* [117,145], the floral homeotic gene *pMADS3* in petunias [146], and the hundreds of genes involved in tomato fruit ripening [147]. Harris et al. (2018) identified a pair of proteins (SUVH1 and SUVH3) that could recognize and bind to methylated DNA. The binding of these proteins is associated with an accessible chromatin state and enhanced gene expression. Upon binding to methylated DNA, SUVH1 and SUVH3 act as recruitment platforms for DNAJ1 and DNAJ2, and together, this protein complex promotes the expression of the genes proximal to their binding site (Figure 5B). More recent works have reported the involvement of a third DNAJ protein (DNAJ3) in the complex, also showing that SUVH 1/3 and DNAJ 1/2/3 are mutually dependent on binding to the methylated sites [148,149].

### 4.2. Regulatory Function of Epialleles in the Gene Body

Gene body methylation (gbM) is common in plants and is typically characterized as the enrichment of CG methylation at the coding regions of a gene. An example of well-studied gbM with a regulatory function can be observed in the collection of epiallele variants of the flower developmental gene *SUPERMAN (SUP),* called the *clark kent (clk)* epialleles. Hypermethylation in the *SUP* gene body results in the abnormal development of floral whorls. Various *clk* variants have been identified with different methylation sites, flower phenotypes, and *SUP* transcriptional levels, suggesting a complex epigenetic regulation of the epiallele [150,151]. Originally, the *clk* lines were identified from the homozygous *ddm1* and *ddm2* mutant populations, but recently, naturally occurring epialleles associated with *SUP* called *lois lane (lol)* have also been identified in 12 different *A. thaliana* accessions [18]. The *clk* epialleles show that gbM can have functional implications, although the methylation pattern at the *SUP* gene body is rather unique, since it is a developmental gene locus that is dominated by CHG and CHH methylation, unlike canonical CG-rich gbM, which is commonly found at constitutively expressed genes. In a recent work, a methylome and transcriptome analysis of 948 *A. thaliana* natural accessions revealed positive but modest associations between canonical CG-rich gbM and the gene expression. These methylation and transcription associations are purely epigenetic and independent of any genetic variations. Further, the epigenome-wide association (epiGWA) analysis showed that some of the gbM were associated with various adaptive traits, such as plant fitness in dry and hot environments, flowering time, and plant adaptability to spring atmospheric NO_2_. The authors of this work suggested the involvement of gbM in gene regulation and adaptive evolutionary processes, although the direct mechanistic evidence of gbM modulation of the gene expression and the associated phenotypes has yet to be defined [152].

Zilberman (2017) suggested that the main function of gbM is likely to be homeostatic, possibly to enhance the mRNA splicing accuracy and efficiency [153] (Figure 5C). In accordance, using single-cell transcriptome data acquired from *A. thaliana* root quiescent center (QC) cells, Horvath et al. (2019) described a lower number of RNA-seq reads mapped to introns in gbM genes compared to unmethylated genes. This result indicates a reduced level of intron retention in the mRNA of gbM gene in quiescent center (QC) cells, suggesting a role for gbM in splicing [154]. Similar results have also been reported in rice where gbM can affect the alternative splicing events. Comparing the WT and *OsMet1* and *2* mutant lines, Wang et al. (2016) described that around 7% of the alternative exon–intron junctions were influenced by a global loss of CG methylation in *OsMet1* and *2*, further suggesting gbM involvement in mRNA splicing [155]. At the single-locus level, it is reported that the loss of methylation at a repeat sequence in the intron of the *DEFICIENS* gene in cloned oil palms caused alternative splicing and premature termination of the gene transcript, resulting in altered fruit phenotype in the plants [103]. Overall, these results suggest the role of gbM in the fine-tuning of mRNA splicing (Figure 5C).

### 4.3. Regulatory Function of Epialleles at Distal Elements

Within the cell, tight chromatin condensation enables long-distance interactions of regions that may be distally located within one chromosome or across various chromosomes. Such chromatin loops create regions of varied accessibility, which could potentially link long-distance enhancer elements to promoter regions and influence the gene expression (Figure 5A,D). *A. thaliana* mutants that are deficient in CG, CHG, or CHH DNA methylation at the genome-wide level have been shown to have increased chromatin accessibility and long-range chromatin interactions, suggesting the regulatory role of DNA methylation on chromatin properties, long-distance chromatin interactions, and gene transcriptions [156].

The first report for enhancer–promoter looping in plants came from the maize *b1* locus. Two *b1* epialleles (*B’* and *B-I*) exist in maize and have identical sequences but show a 10-to-20-fold difference in the expression levels. *B’* and *B-I* carry an enhancer located 100 kb upstream of the *b1* TSS. This enhancer contains seven tandem repeats (hepta-repeat) of an 853-bp sequence that is differentially methylated between the *B’* and *B-I* epialleles. In hypomethylated *B-I*, the enhancer can bind to TFs and a set of protein activation complexes to form a multiloop structure with the *b1* TSS region, promoting H3 acetylation and gene expression. In B’ epialleles, the methylated repeats recruit the protein silencing complex, allowing the formation of only a single-loop structure with the *b1* TSS, promoting the recruitment of the repressive H3K27me2 and H3K9me2 histone marks and subsequently downregulating the gene [157,158] (Figure 5A). Methylation can also affect the enhancer activity in *A. thaliana* when it is artificially introduced by inverted-repeat-hairpins at two enhancers located 5-kb upstream and 1-kb downstream of the *FT* gene, resulting in the downregulation of the gene and deposition of H3K9me2 on the methylated sites [118].

Apart from regulating the enhancer activity, DNA methylation can also modulate the chromatin interactions between two adjacent genes (Figure 5D). In *A. thaliana*, auxin-controlled long noncoding RNA (lncRNA) *APOLO* regulates the expression of the nearby gene *PID* through sequence complementarity and the formation of a chromatin loop. In the absence of auxin, a chromatin loop encompassing *APOLO* and the *PID* promoter region is formed, accompanied by the deposition of DNA methylation and H3K27me3, which leads to the repression of both genes. In response to auxin, this region is demethylated through ROS1, DML2, and DML3 activity, which opens the chromatin loop. The repressive histone mark H3K27me3 is then replaced by the permissive histone mark H3K9Ac, resulting in the activation and transcript accumulation of both genes. The accumulation of the *APOLO* transcript, in turn, recruits LIKE HETEROCHROMATIC PROTEIN 1 (LHP1), RdDM machinery, and PRC2, which facilitate the reformation of the chromatin loop, DNA methylation, and deposition of the repressive histone marks in this region (Figure 5D) [159,160].

DNA methylation-mediated gene regulation is not only restricted to a modulation of the promoter and enhancer activities. Differential methylation at the TEs downstream of a gene can also regulate the gene expression through the control of an antisense lncRNA transcript. The multigenerational exposure to salt stress in *A. thaliana* leads to a heritable loss of DNA methylation at the TEs downstream of *CNI1*, which induces the expression of *CNI1* antisense lncRNA and repression of the CNI1 sense transcript [74]. In maize, Lv et al. (2019) identified 509 TE-derived lncRNAs that were differentially expressed under abiotic stress, with the expression of those TE–lncRNA being significantly correlated to the expression of the proximal genes [161]. However, the functional role of such TE–lncRNA interactions and the involvement of DNA methylation in their regulation remains to be elucidated.

## 5. Conclusions and Perspectives

The study of cytosine methylation marks has demonstrated how gene transcription can be fine-tuned beyond the genetic level of control. Although epigenetic variations might provide transcriptional and phenotypic plasticity, which is beneficial for adaptation, it could also have detrimental effects on the phenotype and fitness of an individual. DNA methylation plays a pivotal role in the suppression of TE mobilization to control highly mutagenic TEs from disrupting the essential genes and ensuring the optimal plant fitness. Even in genotypes where the global methylation levels are altered, the epigenetic homeostasis is faithfully maintained in certain loci involved in the key developmental and regulatory processes, such as the “methylstat” loci [145,162]. Therefore, the re-establishment of DNA methylation and chromatin marks during reproduction is essential in maintaining the integrity of such loci. This reprogramming is important in imprinting and required to ensure an embryo’s survival, as inferred from the developmental defects and inbreeding depression observed in various epigenetic mutants that accumulate epimutations [163,164]. In addition, methylated cytosines are more prone to deamination, and methylated regions are associated with higher mutation rates. Therefore, limiting the epimutation rate is necessary to circumvent deleterious mutations.

Upon collecting the results from several works, we showed that the majority of naturally occurring epialleles in plants are shaped by the underlying structural variations and TE landscapes, suggesting that epiallelic variations may occur as a byproduct of natural selection in genetic variations. However, in the process of adaptation, several “pure epialleles” appeared to be established by chance, independent of the genetic variations, possibly due to errors in the epigenetic reprogramming machinery during the reproductive processes. These newly formed epialleles may interact with other epigenetic features and affect gene regulations only several generations after their initial establishment. It is also possible that they only catalyze a regulatory cascade after a certain environmental cue is experienced. The significance of these pure epialleles in adaptive evolution is still unclear, since the evidence for pure epialleles with biological functions is still limited. Another possible role of such epialleles could be to influence the genetic structural variations for accelerating adaptations. Such a theory has recently been supported by a study showing that cytosine methylation and cytogenetic features can together function as predictors of the de novo genetic mutation rates across protein-coding genes [165].

The majority of the methylation changes induced by environmental stresses, clonal propagation, or artificial epimutagenesis are reset to the basal levels in the non-stressed, and transgene-free progeny. However, some changes can be stably transmitted through many rounds of mitotic and meiotic divisions and are heritable [4,125]. The heritability of the induced epialleles seems to be affected by three factors, the first being the magnitude of the initial epigenetic change. Strong and complete initial methylation changes, introduced across different methylation contexts, such as those introduced by epiRILs, clonal lines, and epimutagenesis lines, appear to be stable and heritable. In contrast, moderate and partial methylation changes, such as those found in plants exposed to environmental stresses, appear to be transient or nonheritable. However, although rare, there are some examples for transgenerational stress-induced epigenetic changes [65,66,81]. The duration, magnitude, and frequency of the applied stresses may influence the heritability of these changes [166]. The second factor affecting epiallele heritability is the genetic landscape of the locus. A low CG content and high repeat copy numbers in a locus are often associated with an increased probability of resetting the methylation marks, while stable and heritable epialleles usually arise from regions with a high CG content and lower repeat density. The third factor involved in the process is the local chromatin environment, where a high level of the euchromatin marks H3K4me3 and H3K18a is associated with the heritability of the epiallelic state.

Methylation patterns at specific loci in the genome may either directly or indirectly (interacting with chromatin proteins) influence an altered gene transcription. It can have a different mode of action depending on its genomic position. At promoters, methylation can repress transcription by recruiting repressive histone marks and inhibiting TF binding. Within gene bodies, it may affect alternative splicing, while, at distal regions, it can affect transcription by regulating the formation of chromatin loops. The availability of molecular tools for engineering targeted methylation changes and the creation of new epialleles is a game-changer for plant epigenetics—this paves the way for characterizing and identifying the evolutionary and functional purposes of numerous methylated regions in the plant genome, either at the individual epiallele level or as a network of epialleles. Overall, identifying the key developmental epialleles, their heritability, and their modes of action on transcription are instrumental for engineering plants with improved fitness, yields, and tolerance to stresses.

## Figures and Tables

**Figure 1 ijms-22-08618-f001:**
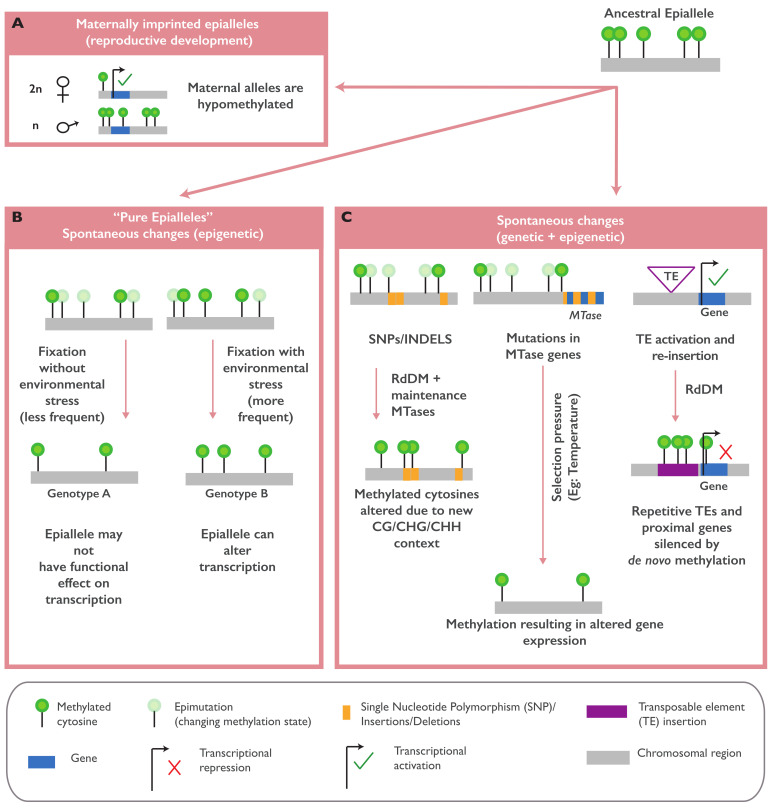
The origins of natural epialleles. Imprinted epialleles can be generated during the reproductive development in wild-type (WT) plants (**A**) where parental allele-specific methylation marks are removed, initiating gene transcription. Spontaneous methylation changes can occur independent of the underlying genetic variation (“pure epialleles”) and may also be subjected to evolutionary fixation (**B**). Mutations in protein-coding genes, including genes involved in the maintenance of DNA methylation (methyltransferases or “MTases”) and the mobilization of transposable elements (TEs), can also recruit methylation marks, which result in the formation of epialleles, as they can influence the proximal gene transcription (**C**). Together, these newly acquired genetic and epigenetic changes may provide an adaptive advantage for the plant.

**Figure 2 ijms-22-08618-f002:**
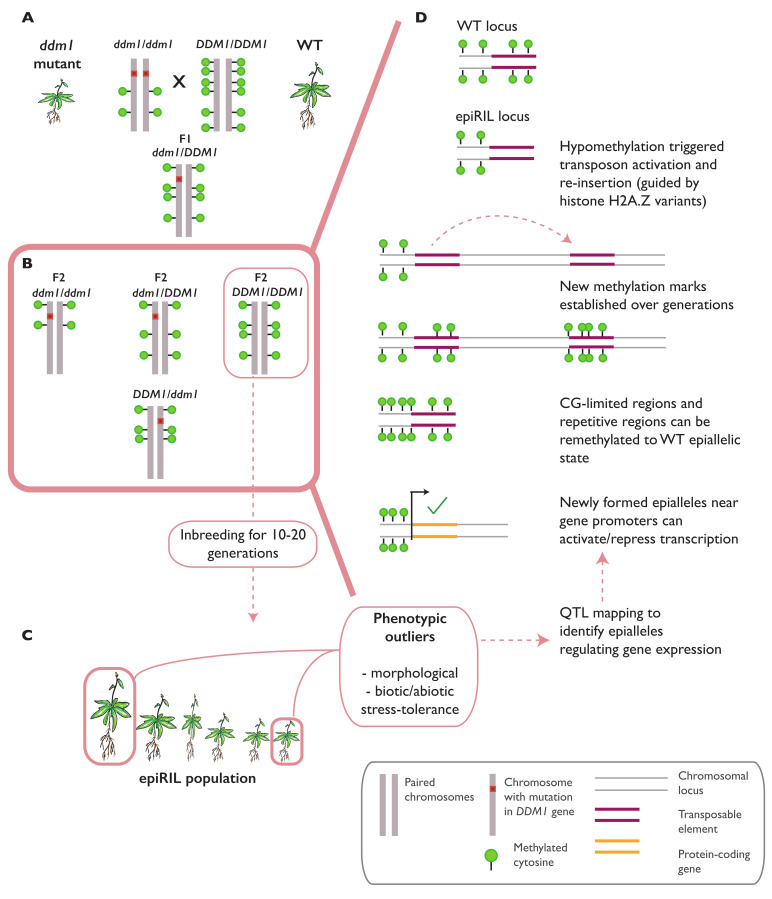
Identification of novel epialleles by epiRIL generation. Genome-wide methylation patterns can be redistributed by crossing wild-type (WT) plants with mutants defective in DNA methylation (such as the *ddm1* mutant) (**A**). Progeny from such crosses carrying the WT alleles (DDM1/DDM1) are selected for multigenerational inbreeding (**B**), ultimately generating epigenetic recombinant inbred lines (epiRILs). The epiRIL population can be tested for variations in several morphological or stress resistance traits to identify the phenotypic outliers (**C**). These lines are ultimately chosen for an epigenetic quantitative trait loci (epiQTL) analysis to identify the novel epialleles underlying the measured traits. Such epialleles in epiRIL lines can occur due to the activation of transposable elements (TEs) and their subsequent reinsertion into distal loci, often determined by chromatin properties and the nature of the target sequence (CG content) (**D**).

**Figure 3 ijms-22-08618-f003:**
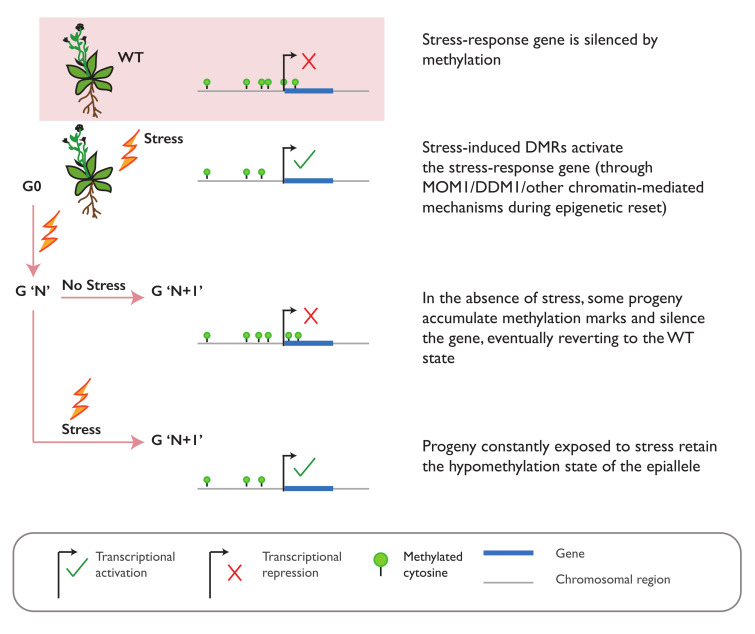
Epiallele formation upon exposure to stress. A stress response gene that is silenced by DNA methylation in wild-type (WT) plants can be activated upon stress exposure by mechanisms that remove the methylation marks. These stress-induced differentially methylated regions (DMRs) may occur immediately in the first generation (G0) but may be heritably carried over only upon repeated exposure to stress (generations G’N’ and G’N+1’). This stress-induced epigenetic memory in the progeny can eventually be released once the stress is alleviated, reverting the epiallele to its WT methylation state.

**Figure 4 ijms-22-08618-f004:**
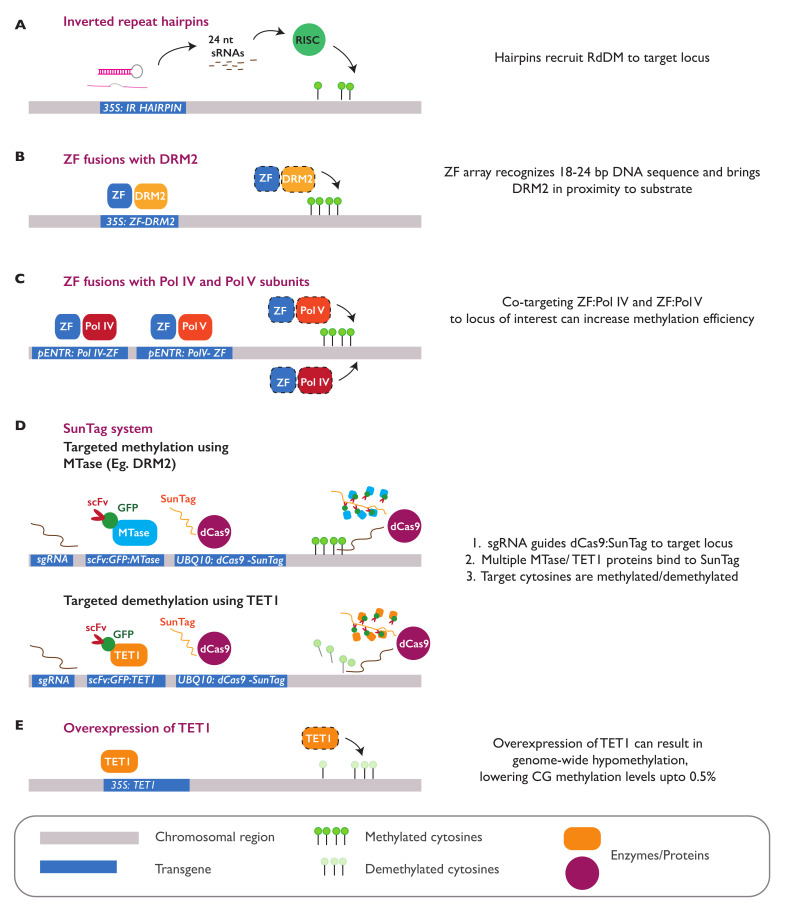
Different methods for targeted epigenetic engineering. DNA methylation can be introduced into specific regions in the genome using IR-hairpin constructs that can recruit a sRNA-induced silencing complex (RISC) into the target sequence (**A**), ZF constructs fused with methyltransferase genes such as *DRM2* that can directly methylate cytosines at a target sequence through RdDM pathways (**B**), and ZF constructs fused with genes coding for Pol IV and Pol V subunits working in tandem to recruit RdDM machinery into the target sequence (**C**). Combining dCas9 with the SunTag system is another molecular engineering approach that allows the multimerization of multiple methyltransferases (“MTases”) (e.g., DRM2) or demethylases (e.g., TET1), which can selectively methylate or demethylate the target sequence, respectively (**D**). The SunTag system comprises two modules—the first being dCas9 and an epitope tail and the second being a single chain antibody (scFv) fused to GFP and the MTase of interest. The removal of genome-wide DNA methylation can also be catalyzed by overexpression of the *TET1* gene (**E**).

**Figure 5 ijms-22-08618-f005:**
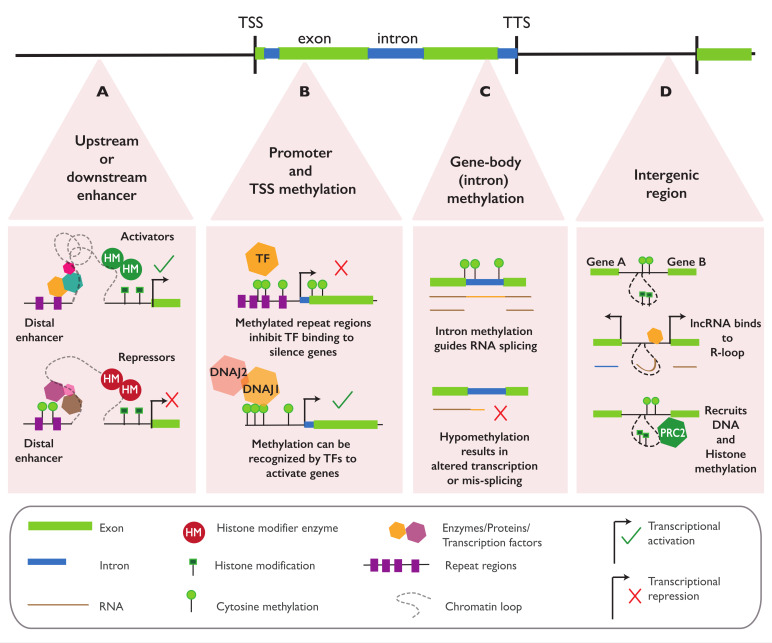
Different mechanisms for gene regulation by epialleles. In the vicinity of a gene, methylation can occur in several regions, with a distinct function in each. In regions upstream or downstream of the genes that house enhancer sequences, the presence of DNA methylation can promote the formation of a single-loop structure to recruit repressive histone marks, causing the downregulation of the associated gene. The removal of methylation in the distal enhancer can promote transcription factor (TF) binding, the recruitment of permissive histone marks, and the formation of a multiloop structure, which, together, promote the expression of the associated gene (**A**). At the promoter and transcription start-site (TSS) of a gene, DNA methylation (often at repeat-rich regions) can inhibit TF binding and recruit repressive histone marks that, in turn, repress or completely silence the transcription of the associated gene. Depending on the sequence and methylation context, DNA methylation can also have a positive effect on the gene expression; it can recruit protein complexes (such as DNAJ1 and DNAJ2) that promote the expression of the genes proximal to methylated sites (**B**). In the gene body, DNA methylation may be involved in the regulation of intron-splicing in mRNA. The loss of methylation in the intron can lead to the mis-splicing and premature transcript termination of the associated gene (**C**). DNA methylation can also modulate chromatin interactions between two adjacent genes, together with a repressive histone mark promoting the formation of a loop structure encompassing the two adjacent genes and repressing both genes. Upon the removal of methylation in the region, the chromatin loop is opened, and permissive histone marks replace the existing marks, resulting in the activation of both genes. In a self-feedback mechanism, the excessive accumulation of gene transcripts leads to the generation of lncRNA, reformation of the chromatin loop, hypermethylation, and the deposition of repressive histone marks (by proteins such as PRC2) in this region (**D**).

## Data Availability

Not applicable.

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
