# Peer review of "The Underlying Nature of Epigenetic Variation: Origin, Establishment, and Regulatory Function of Plant Epialleles"

_ijms, 2021, doi:10.3390/ijms22168618_

Round 1

Reviewer 1 Report

It is an informative review on the functional aspects of epigenetic variation in plants, generally well-written and clearly presented.

I suggest to add a brief information on the different cytosine methylation contexts (CG, CHG, CHH) early in the introduction, with appropriate references, as the term 'non-CG methylation' appears in line 134 without any prior explanation.

Also, it would be useful to add comprehensive legends to figures informaing about the meaning of the graphical symbols used to depict the described phenomena (light and dark green circles, lines of different colors, etc.).

In par. 4.2 on gene body methylation, I believe that using SUP as 'a classic example' is slightly misleading; as a rule gbM refers predominantly to stable levels of CG methylation in constitutively expressed genes, while dynamic changes in non-GC gbM seem to be rather an exception...

Minor:

from par. 3.2 onwards, latin names and gene symbols are not italicized in the text, it should be corrected

line 122: L. vulgaris - the species name should not be abbreviated it is the only instance it appears.

Author Response

1. It is an informative review on the functional aspects of epigenetic variation in plants, generally well-written and clearly presented.

Response:  Thank you for your encouraging words, we are pleased to know that the paper was observed to be informative, well-written, and clearly presented.

2. I suggest to add a brief information on the different cytosine methylation contexts (CG, CHG, CHH) early in the introduction, with appropriate references, as the term 'non-CG methylation' appears in line 134 without any prior explanation.

Response: Thank you for the suggestion.  We have added a paragraph in the Introduction to briefly introduce different cytosine methylation contexts in plants (line 32 to 41 in the revised manuscript).

3. Also, it would be useful to add comprehensive legends to figures informing about the meaning of the graphical symbols used to depict the described phenomena (light and dark green circles, lines of different colors, etc.).

Response: Thank you for the suggestion. We have now revised all of the figures in the manuscript according to suggestions from both reviewers. We have added comprehensive legends at the bottom of every figure, explaining all the symbols used and the color scheme in detail. We have also split Figure 2 into two figures (now Figure 2 and Figure 3), for better clarity.  Additionally, we have enlarged all figures, and also increased the font size of all labels to improve the readability.

4. In par. 4.2 on gene body methylation, I believe that using SUP as 'a classic example' is slightly misleading; as a rule gbM refers predominantly to stable levels of CG methylation in constitutively expressed genes, while dynamic changes in non-GC gbM seem to be rather an exception... 

Response: Thank you for correction, we agree that SUP is not “a classic example” of gbM, although it is located in the gene body the methylated regions of SUP is dominated by dynamic CHG and CHH methylation, unlike stable CG commonly found in canonical gbM. We have revised the SUP section of the manuscript to make this point clearer.

Line 627-629: An example of well-studied gbM with regulatory function can be observed in the collection of epiallele variants of the flower developmental gene SUPERMAN (SUP), called the clark kent (clk) epialleles. 

Line 637-641: The clk epialleles show that gbM can have functional implications, although the methylation pattern at the SUP gene body is rather unique since it is a developmental gene locus which is dominated by CHG and CHH methylation, unlike canonical CG-rich gbM which is commonly found at constitutively expressed genes.

5. From par. 3.2 onwards, latin names and gene symbols are not italicized in the text, it should be corrected

Response: We apologize for incorrectly formatting some of the gene and species names, we have revised the manuscript accordingly.

6. Line 122: vulgaris - the species name should not be abbreviated it is the only instance it appears.

Response: The full name Linaria vulgaris already mentioned in line 61, which is why it is abbreviated in line 133.

Reviewer 2 Report

Review of IJMS - 1325165

The Underlying Nature of Epigenetic Variation: Origin, Establishment, and Regulatory Function of Plant Epialleles

by Srikant and Wibowo

The topic of the nature of epigenetic variation is of great interest to both expert and general readers (non-experts in epigenetics). I have a couple of general comments that could improve the readability of this review for the general reader. I have also provided and itemized list of corrections and suggestions that should help the authors make their paper clearer.

1) A brief introductory section in which the basic terms of epigenetics are explained would greatly help the general reader to get a better understanding of the main topics of the paper. This could be as a box, box and figure, or just a couple of paragraphs that introduce some of the symbology and terminology used in the text and figures. Alternatively, the authors could point the reader to one or two reviews of epigenetics that would be helpful in achieving the basic understanding of epigenetics needed to get a good grasp of their paper.

2) The last couple of paragraphs address the proximal causes of epiallele stability / heritability. However, the main reason one could suspect to underly the tendency to revert to a “basal level” is a cost of the changed methylation states or patterns. Adding a brief discussion of the possible costs of different methylation patterns would provide a more complete view of the factors that may influence epiallele heritability, and might point to some limitations regarding the use of epigenetic in the engineering of plants.

Other points:

The abstract defines Differential Methylated Positions, but these are not discussed further in the main text. They should be dropped from the abstract or explained / discussed in the main text, perhaps in contrast to the DMRs.

L54: change “independent” to “regardless”

L55: It is not clear what the authors mean by “natural evolutionary process” in contrast to the stimuli and treatments listed, which could be involved in evolutionary processes in natural populations. Perhaps it would be clearer if the authors use the expression that appears in the next paragraph: “naturally occurring” versus “artificially induced” epialleles

L80: Consider replacing “against” with “to”

L82: If spontaneous epialleles” is the name given to the stable and heritable epialleles that arise from spontaneous methylation changes, this needs to be explicitly stated at this point: consider editing as follows “… and heritable epialleles ‘“spontaneous alleles’), although…”

L101: change “constitute” to “consecutive”

L127: This sentence is not informative: the detection of a high correlation between any two variables indicates a tight association between the two variables. The most important idea is what follows: an interpretation of the biological reasons for the association found between the methylation and genetic variation. One suggestion is to change the second clause of the sentence: “…suggesting a tight genetic control of methylation.”  (The use of “tight” is my own interpretation).

L131: Are there any references for this statement.

L185: use the singular form “variation”

L195: as above: “replication”

L198: the term “functional epiallele” has been used several times. Are there non-functional epialleles? Please explain at the first use of the expression.

Figure 1 is too small, difficult to read. Symbology needs to be explained. The nomenclature of the panels is confusing: aren’t there three panels only? top, bottom left and bottom right?

L229: remove “inherited”

L232: “allele-free generations” needs to be explained

L241: use italics for species name

L255: italics needed;  check throughout the text henceforth

L274: replace “reduce” with “decrease”

L280: misuse of “constitutive”

L290: remove “that”, and consider removing “often”

L295: switch to “oil palm” , see also L301 and henceforth along the text

L313: This is quite interesting. Would the pattern of epigenetic changes be related to the epigenetic events that occur during the development of the different organs?

Figure 2 also too small. Consider splitting into two figures to make everything larger.

Figure 3: too small

L357: check use of italics

Figure 4: enlarge; consider naming the panels with letters a-d so as to make it easier to refer to them

L590-596: It seems just as likely that new epialleles would have a detrimental effect on the phenotype, and therefore, on the fitness of an individual.

Author Response

1. The topic of the nature of epigenetic variation is of great interest to both expert and general readers (non-experts in epigenetics). I have a couple of general comments that could improve the readability of this review for the general reader. I have also provided and itemized list of corrections and suggestions that should help the authors make their paper clearer.

Response: Thank you for the positive feedback, we appreciate your comments, corrections, and suggestions throughout the text, all of them were greatly beneficial for improving the readability and quality of this manuscript. We have addressed those comments in the following responses.

2. A brief introductory section in which the basic terms of epigenetics are explained would greatly help the general reader to get a better understanding of the main topics of the paper. This could be as a box, box and figure, or just a couple of paragraphs that introduce some of the symbology and terminology used in the text and figures. Alternatively, the authors could point the reader to one or two reviews of epigenetics that would be helpful in achieving the basic understanding of epigenetics needed to get a good grasp of their paper.

Response: Following reviewer suggestion we added a “glossary box” near the introduction. We hope this will help the general reader to better understand our review and the terminology we have used throughout the manuscript. 

3. The last couple of paragraphs address the proximal causes of epiallele stability / heritability. However, the main reason one could suspect to underly the tendency to revert to a “basal level” is a cost of the changed methylation states or patterns. Adding a brief discussion of the possible costs of different methylation patterns would provide a more complete view of the factors that may influence epiallele heritability, and might point to some limitations regarding the use of epigenetic in the engineering of plants.

Response: This is a good point since we have not discussed the negative consequences of accumulating epimutations on plant fitness. We have now added a paragraph under Conclusions and Perspectives to discuss the possible cost of epimutations and why it is necessary for plants to limit their epimutation rate.

In Line 770-784: Although epigenetic variation might provide transcriptional and phenotypic plasticity that is beneficial for adaptation, it could also have detrimental effects on the phenotype and fitness of an individual. DNA methylation plays a pivotal role in the suppression of TE mobilization, to control highly mutagenic TEs from disrupting essential genes and ensuring optimal plant fitness. Even in genotypes where global methylation levels are altered, epigenetic homeostasis is faithfully maintained in certain loci involved in key developmental and regulatory processes, such as the “methylstat” loci [145,162]. Therefore, the re-establishment of DNA methylation and chromatin marks during reproduction is essential in maintaining the integrity of such loci. This reprogramming is important in imprinting and required to ensure the embryo's survival, as inferred from developmental defects and inbreeding depression observed in various epigenetic mutants that accumulate epimutations [163,164]. In addition, methylated cytosines are more prone to deamination and methylated regions are associated with higher mutation rate. Therefore, limiting epimutation rate is necessary to circumvent deleterious mutations

4. The abstract defines Differential Methylated Positions, but these are not discussed further in the main text. They should be dropped from the abstract or explained / discussed in the main text, perhaps in contrast to the DMRs.

Response: Thank you for pointing this out. We have deleted the term “Differential Methylated Positions” from the abstract since we are not discussing it in the main text.

5. L54: change “independent” to “regardless”

Response: We have made the suggested change.

6. L55: It is not clear what the authors mean by “natural evolutionary process” in contrast to the stimuli and treatments listed, which could be involved in evolutionary processes in natural populations. Perhaps it would be clearer if the authors use the expression that appears in the next paragraph: “naturally occurring” versus “artificially induced” epialleles

Response: In accordance with the above suggestion we have changed “natural evolutionary process” to “naturally occurring” and contrasted it with “artificially induced” epialleles.

Line 67-71: Besides naturally occurring in the population, epialleles can also be artificially induced by wide-range of experimental stimuli and treatments. They might arise following biotic or abiotic stress [4], chemical treatments [27,28], clonal propagations [29], interference of methylation pathways [30,31], and/or due to deficiencies in the maintenance of their chromatin states [31,32].

7. L80: Consider replacing “against” with “to”

Response: We have made the suggested change.

8. L82: If spontaneous epialleles” is the name given to the stable and heritable epialleles that arise from spontaneous methylation changes, this needs to be explicitly stated at this point: consider editing as follows “… and heritable epialleles ‘“spontaneous alleles’), although…”

Response: We have edited the text according to reviewer suggestions.

Line 100-102: A more recent study of the same line confirmed that spontaneous methylation changes can lead to the formation of stable and heritable epialleles (“spontaneous epialleles”), although occurring at a very low frequency

9. L101: change “constitute” to “consecutive”

Response: We made the suggested change.

10. L127: This sentence is not informative: the detection of a high correlation between any two variables indicates a tight association between the two variables. The most important idea is what follows: an interpretation of the biological reasons for the association found between the methylation and genetic variation. One suggestion is to change the second clause of the sentence: “…suggesting a tight genetic control of methylation.”  (The use of “tight” is my own interpretation).

Response: Following this suggestion, we have paraphrased the sentence to make it clearer for reader.

Line 149-152:  Instead, evidence for linkage and strong association between methylation and genetic variation has been observed in inbred lines of soybean [36], B. distachyon [41,43], maize [39], and A. thaliana natural populations [9,44–46], suggesting that the majority of methylation variation found within a species could have a genetic basis.

11. L131: Are there any references for this statement.

Response: We added the appropriate references into the statement. 

12. L185: use the singular form “variation”

Response: We apologize for the grammatical mistake, we have revised the manuscript accordingly.

13. L195: as above: “replication”

Response: We have revised the manuscript accordingly.

14. L198: the term “functional epiallele” has been used several times. Are there non-functional epialleles? Please explain at the first use of the expression.

Response: From our understanding, any heritable variation or alternative forms of epigenetic marks within a species can be called epiallele. There are numerous epialleles identified in Arabidopsis thaliana (mostly as DMRs) but only a small fraction of them have been assigned biological functions. Therefore, we use the term “functional epiallele” to describe epialleles that have a known regulatory function, to differentiate it with those that have no known biological function. To make this clear we add following explanation when we first mention “functional epiallele” in the text:

Line 80-82: In this review we will discuss different mechanisms and genomic features that could facilitate the establishment of functional epialleles (epialleles with an assigned biological role) in a plant’s genome………..

15. Figure 1 is too small, difficult to read. Symbology needs to be explained. The nomenclature of the panels is confusing: aren’t there three panels only? top, bottom left and bottom right?

Response: Thank you for the suggestion. We have now revised all of the figures in the manuscript according to suggestions from both reviewers. We have added comprehensive legends at the bottom of every figure, explaining all the symbols used and the color scheme in detail. We have also split Figure 2 into two figures (now Figure 2 and Figure 3), for better clarity.  Additionally, we have enlarged all figures, clearly defined sub-panels (A,B,C,etc.) and also increased the font size of all labels to improve the readability.

16. L229: remove “inherited”

Response: We made the suggested change.

17. L232: “allele-free generations” needs to be explained

Response: We paraphrase the sentence to clarify what we means by “allele-free generations”

Line 276-279:  In ddm1-2 epiRILs, such re-methylation can occur progressively across one to three generations after the F1 hybrid is  selfed or reciprocally backcrossed with the WT to produce individuals that lack the mutant allele (DDM1/DDM1). 

18. L241: use italics for species name

Response: We apologize for incorrectly formatting species names, we have revised the manuscript accordingly. 

19. L255: italics needed; check throughout the text henceforth

Response: As mentioned, apologize for incorrectly formatting gene symbols and species names, we have revised the manuscript accordingly. 

20. L274: replace “reduce” with “decrease”

Response: We have made the suggested change.

21. L280: misuse of “constitutive”

Response: We delete “constitutive” from the sentence

22. L290: remove “that”, and consider removing “often”

Response: We have made the suggested change.

23. L295: switch to “oil palm” , see also L301 and henceforth along the text

Response: We apologise for the wrong use of the word, we have changed “palm oil” to “oil palm” for the entire text.

24. L313: This is quite interesting. Would the pattern of epigenetic changes be related to the epigenetic events that occur during the development of the different organs?

Response: Roots and leaves have different methylation signatures, further distinguished by their cellular identities where each cell type has their own methylation pattern. Root and leaf identity are established very early during embryonic development, however the epigenetic regulation and the importance of DNA methylation in this process is unclear since there is no methylome data available to date from apical and basal cells of the early embryo. Although each organ or even cell has their own specific methylation, their importance is unclear since different epimutants in A.thaliana such as met-1, ddm1, and nrpd1a  are not severely impaired in their organ development and produce proper roots and leaves, although several developmental abnormalities during seed development and flowering are associated with DNA methylation. In other plants with more complex genomes, DNA methylation might serve a more crucial role in organ development, for example in maize where defects in regulation of methylation, whether it is CG or non-CG could lead to sterility or embryo lethality. 

We did not discuss this in our review since we are not focusing on the role of DNA methylation in plant development and rather focusing more generally on the regulation of gene expression by methylation. 

25. Figure 2 also too small. Consider splitting into two figures to make everything larger.

Response: Thank you for the suggestion, as mentioned we have split Figure 2 into two figures (now Figure 2 and Figure 3), for better clarity.  Additionally, we have enlarged all figures, clearly defined sub-panels (A,B,C,etc.)  and also increased the font size of all labels to improve the readability.

26. Figure 3: too small

Response: We have enlarged all figures, and also increased the font size of all labels to improve the readability.

27. L357: check use of italics

Response: We apologize for the incorrect formatting, we have revised the manuscript accordingly.

28. Figure 4: enlarge; consider naming the panels with letters a-d so as to make it easier to refer to them

Response: We have enlarged the figures and clearly defined sub-panels (A,B,C,etc.)  as suggested. 

29. L590-596: It seems just as likely that new epialleles would have a detrimental effect on the phenotype, and therefore, on the fitness of an individual.

Response

We agree, newly established epialleles can have beneficial or detrimental effects on plant fitness, depending on the type and location. We make this clear in the following  paragraph (as mentioned previously):

In Line 770-784: Although epigenetic variation might provide transcriptional and phenotypic plasticity that is beneficial for adaptation, it could also have detrimental effects on the phenotype and fitness of an individual. DNA methylation plays a pivotal role in the suppression of TE mobilization, to control highly mutagenic TEs from disrupting essential genes and ensuring optimal plant fitness. Even in genotypes where global methylation levels are altered, epigenetic homeostasis is faithfully maintained in certain loci involved in key developmental and regulatory processes, such as the “methylstat” loci [145,162]. Therefore, the re-establishment of DNA methylation and chromatin marks during reproduction is essential in maintaining the integrity of such loci. This reprogramming is important in imprinting and required to ensure the embryo's survival, as inferred from developmental defects and inbreeding depression observed in various epigenetic mutants that accumulate epimutations [163,164]. In addition, methylated cytosines are more prone to deamination and methylated regions are associated with higher mutation rate. Therefore, limiting epimutation rate is necessary to circumvent deleterious mutations.
